# Perception of the quality of remote lessons in the time of COVID-19: A comparative study in Latin America quality of remote lessons in the context of COVID-19

**Lida Esperanza Villa Castaño**[1]☯*, **William Fernando Durán**[2]☯, **Paula Andrea Arohuanca Percca**[3]

1 Faculty of Economics and Administrative Sciences, Department of Business Administration, Universidad Cooperativa de Colombia, Sede Bogotá, Bogotá D.C, Colombia, 2 Faculty of Economics and Administrative Sciences, Department of Business Administration, Pontificia Universidad Javeriana, Bogotá, Colombia, 3 Faculty of Accounting and Administrative Sciences, Universidad Nacional del Altiplano de Puno-Perú, Puno, Perú

☯ These authors contributed equally to this work.
* Lida.villacas@campusucc.edu.co

**Data Availability Statement:** All relevant data are within the manuscript. The database is available and open to the public in the repository of the

## Abstract

This research examines the perception of undergraduate students of public and private universities in Latin America on the quality of the lessons that applied the emergency remote teaching (ERT) in the time of COVID-19. This study employs a mixed sequential approach, starting with six focal groups, and finishing with a quantitative validation exercise that uses exploratory and confirmatory factor analysis as well as regression models. Findings reveal that student perception is elicited along three dimensions: concerns related to academic quality, teaching strategies applied by professors, and access limitations. Moderation analysis shows that the relationship between teaching strategies and the concerns related to academic quality varies and that it even gets stronger when access limitations are reduced. Consequently, perception is limited by student access to maintain the teaching–learning process.

## Introduction

On March 11, 2020, the World Health Organization (WHO) informed the world that COVID-19 had become a pandemic. The health situation alarmed governments and various sectors of economy [1]; the virus had turned into a threat to health and the ordinary course of economic, social, and production business in every corner of the world. Countries gradually started to turn off their production engines to safeguard life as there was also evidence indicating that the weak healthcare system did not allow for maintaining the regular course of social and productive activities [2–4]. In response to this situation, as a public policy, mobility was constrained.

Universidad Cooperativa de Colombia library, and it can be consulted in the following link: https://repository.ucc.edu.co/handle/20.500.12494/43671 or through doi: https://doi.org/10.16925/20.500.12494/43671.

**Funding:** The author(s) received no specific funding for this work.

The challenge for the various sectors of economy involved adapting or reinventing themselves to meet the demands of a society that had to keep on functioning for the imperative of preserving individuals' integrity, and the education sector was no exception. Higher education institutions (HEIs) providing classroom-based training throughout the world were authorized to replace in-person classes for emergency remote teaching (ERT) [5].

ERT became the solution to a temporary change in the provision of education according to the emergency context triggered by the pandemic. The aim was to teach in a simultaneous manner although without a solid teaching and assessment ecosystem [5]. ERT cannot be comparable with online education, also known as remote learning, characterized by the students' flexibility toward meeting their learning goals by, for instance, choosing their study times [6].

In Latin America, presence-based universities resorted to the ERT emerging model, and on March 22, amid all difficulties and improvisations, thousands of students were called to resume their lessons to learn via ERT. The HEIs with financial muscle used paid platforms, while universities with smaller budgets employed free platforms [7]. Nonetheless, in addition to the communication platforms, ERT uncovered the existent issues associated with the technology gap [8–10] and the utopian idea of education as a public good [11]. On the other hand, the sudden change to ERT exceeded the capacity of institutional resources oriented toward in-person work [12].

To ensure pedagogical continuity, the hurried call to resume the academic semester revealed the compelling need to keep the university institution open [13]. However, this disruptive situation raised important questions for society and in-person HEIs, with the most essential being how to educate in times of confinement and in the absence of any historical references. The dominant concern was to accurately preserve the quality of higher education, which has posed a challenge for Latin America [14–17] and worsened after the pandemic in terms of quality and equity [18].

Across the region, all in-person activities taking place in HEIs were quickly and temporarily shut down. Colombia, Peru, Mexico, and Costa Rica closed their campuses on March 12 [18]. Ten days later, ERT was adopted by most public and private universities. The preparation time for both students and professors was minimal, which led to chaos involving a combination of different types of hurdles. Internet access, inexperience in the use of platforms, lack of familiarity with online learning tools and methods among professors and students, lack of resources to access classes from a computer or a smartphone, in addition to unsuitable learning environments, have each played a part [18].

Multiple research studies have been conducted on education in times of COVID-19, adopting various approaches. Attitude toward the use of communication platforms [19–21], perception of learning and commitment [22], dropouts and tuition fees [23], availability of technological resources (McMurtrie, 2020), and professor and student perceptions of online lessons [24] are some problems addressed in economically developed countries and to a lesser extent in developing countries.

In developing countries, and particularly in Latin America, research has implemented three different approaches: technological challenges [25], role of the professor during a pandemic [26], and educational challenges and opportunities [27].

In regard to the perception of students towards class quality in ERT, there are few research studies in the Latin American context. Pérez-Villalobos [12] develop a satisfaction scale on remote teaching for medicine students in Chile. They found that satisfaction is related to teaching contribution, context diagnosis, student participation, and organized relations. On the other hand, Keržič, D., Alex, J. K [28] in his research, which includes students from Mexico and Ecuador, found that perception of students in regards to ERT is connected to the quality of technical-administrative service; a prompt response from professors as well as from

technological infrastructure. Aristovnik et al [28] in a study which took into account countries from the Latin American region concluded that the context of students and access to technological equipment are influential factors in the perception of education quality during ERT.

Although the ability of HEIs to adapt to the uncertainty caused by the COVID-19 pandemic was tested immediately, there is a gap in the literature regarding whether the quality of ERT in the course of the pandemic affected the training process of students in Latin American countries. In this respect, the scope of this research is to analyse the perception of undergraduate students in regards to the quality of remote lessons throughout COVID 19 in Colombia, Costa Rica, Mexico and Peru.

Students' perception of ERT quality is vital as students constitute the fundamental stakeholder of the university institution's risk matrix. Thus, research is useful for universities inasmuch as the results help make future decisions. In a world likely to collapse because of environmental sustainability issues, going back home to study is probably redundant.

Research was conducted using a mixed methodology. In the qualitative phase, six focus groups were created, comprising undergraduate students of various areas from private and public universities located in three important cities of Colombia. These focus groups' research question revolved around the perception of lessons quality that applied ERT.

The findings of the qualitative phase helped to write items using a Likert scale. The pool of items was reviewed by four international peers. The survey was virtually administered from May to August of 2020 in public and private universities of Colombia, Peru, Mexico, and Costa Rica. Results show that student perception in terms of ERT quality is divided into three dimensions: concerns related to academic quality, teaching strategies applied by the professors, and access limitations.

The sections below address the methodology that evidences the mixed design stages, followed by the samples selected for the qualitative and quantitative phases. Next comes a detailed description of the process of building of the quantitative instrument, followed by analysis of results, and then by a discussion demonstrating the theoretical and practical contributions made by the study as well as its limitations.

## Material and methods

A sequential mixed methods design was applied by means of a qualitative study first, followed by a quantitative analysis [29]. This type of design allows one to explore under researched problems and delve into them using measurements and quantitative analyses [30].

### Procedure

In the qualitative phase, six focus groups were done in Colombia via Skype. The requirements to participate in the focus groups were two: Participants had to be 18 years or older and enrolled in an undergraduate program. In the quantitative phase, the researchers designed an instrument to measure the perception of the quality in remote education amidst the pandemic. Items of the instrument were based in the findings of qualitative phase.

Initially this research project was addressed by Universidad Cooperativa de Colombia and Universidad Nacional del Altiplano de Puno in Peru. Nonetheless, thanks to the support of Red Innova Cesal, the Asociación Mexicana para la Internacionalización de la Educación Superior (AMPEI), and the Asociación Nacional de Universidades e Instituciones de Educación Superior (ANUIES), the project was applied in Mexico and Costa Rica. In each country, researchers took into account the legal requirements for capture and analyze data.

The instrument was designed in Google Forms and the link was emailed to institutional mails of students from different undergraduate programs, by professors and administrative

staff from public and private universities in Colombia, Costa Rica, Mexico and Peru. The first part of the instrument indicated the scope of the research, its academic nature, and the informed consent was requested. Participants answered the instrument voluntarily and no personal information such as names, identification numbers, email addresses or phone numbers were collected.

### Ethical considerations

The research committee from the Faculty of Economics and Administrative Sciences of Universidad Cooperativa de Colombia, Sede Bogota approved the research project as the responsible institution. The approval number of the project is 3044.

### Qualitative inquiry

The qualitative study was conducted using six focus groups of undergraduate students from both public and private universities in Colombia. However, in the quantitative phase it is shown that the results in relation to the construct are the same for all countries. Table 1 summarizes the characteristics of the participants in each focus group.

Focus groups were simultaneously conducted via Skype 2 months after the beginning of the pandemic and addressed the following question: How does emergency remote education affect the quality of classes? Each focus group lasted between 50 and 75 minutes. The information was processed through matrixes in Excel, and three categories were identified: 1) 1) technology gap and availability of technological equipment, 2) quality of lessons, and 3) pedagogical tools. To guarantee reliability, a triangular analysis of the data was conducted by the researchers.

### Quantitative inquiry

Thirty-two Likert items were constructed on the basis of the results of the qualitative study, including five categories, with 1 being 'Strongly disagree' and 5 being 'Strongly agree.' These were sent to four judges, including two in-person education experts and one specialist in remote learning, while the remaining judge specialized in the measurements involved in social and behavioral sciences. For evaluating the items, they were provided with the definition of quality education suggested by UNESCO [11], described as "the combination of conditions for the teaching–learning process and the student's academic achievements' (p. 9), to be taken as a reference for them to assess its pertinence and relevance.

**Table 1. Focus groups' sample.**

| | Focus group | | | | | |
|---|---|---|---|---|---|---|
| | **1** | **2** | **3** | **4** | **5** | **6** |
| University | Public | Private | Private | Private | Private | Public |
| Courses | Industrial Engineering, Administration, Sociology and Philosophy | Industrial Design, Social Communication, Administration | International Business and Psychology | Business Administration, Accounting and Economy | Social and Human Sciences, Architecture | Bachelor's degrees in Literature, Biology, Philosophy |
| Average age | 19 years old | 21 years old | 21 years old | 23 years old | 20 years old | 19 years old |
| Semesters | First to fourth | Seventh to ninth | Fifth to seventh | Fifth and sixth | Seventh to tenth | First to third |
| No. of men | 2 | 1 | 4 | 4 | 3 | 4 |
| No. of women | 4 | 5 | 3 | 1 | 5 | 2 |

The comments made by the judges allowed for improving clarity in wording and sought to facilitate the participants' understanding. No item was withdrawn by the judges, so the 32 items constructed were kept, including the suggested changes.

Items were applied online and in parallel in the four countries that participated in this study. The survey was administered to 1,635 students in Colombia, 1,010 in Mexico, 1,218 in Peru, and 182 in Costa Rica. Given the high variability of courses reported by students, these were grouped into knowledge areas in accordance with the classification provided by UNESCO [11]. Table 2 shows the sample description.

Items were grouped through an exploratory factor analysis (EFA) and a confirmatory factor analysis (CFA). Each of these groups was conceived as a dimension of the construct of perception of remote learning quality. In addition, the reliability of each dimension was studied using Cronbach's alpha, which must be above 0.7 to indicate adequate reliability [31]. EFA is a statistical technique that delineates the underlying structure to a set of items, better known as factors, among the variables under study. Moreover, CFA allows for testing out the aforementioned underlying structure [31]. As EFA and CFA must be applied to different samples, the samples collected in Colombia, Mexico, and Peru were randomly divided into two groups, with EFA applied to the first group and CFA applied to the second. Only EFA was applied to Costa Rica as the sample size did not allow for obtaining a sufficient number of cases to conduct both analyses.

EFA was performed independently with data from each country, and their results were compared. The polychoric correlation matrix was used, given the ordinal nature of the Likert items [32], in addition to oblique rotation with the unweighted least squares (ULS) estimate method [31]. A parallel analysis was conducted to establish the number of factors to be retained [33]. Furthermore, items weighing less than 0.4 in at least two countries were removed in accordance with the criteria proposed by Hair et al. [31] for the minimum factorial weight. CFA also implemented polychoric correlations, and the model adjustment was

**Table 2. Sample description.**

| Variable | | Colombia | México | Perú | Costa Rica |
|---|---|---|---|---|---|
| Gender | Women | 57,98% | 59,11% | 63,46% | 40,11% |
| | Men | 41,41% | 40,79% | 36,54% | 59,34% |
| | Other | 0,61% | 0,10% | 0,00% | 0,55% |
| Age | Under 18 | 7,03% | 16,83% | 3,04% | 6,04% |
| | 19–25 | 71,87% | 47,72% | 88,10% | 85,16% |
| | 26–30 | 13,82% | 31,78% | 6,90% | 3,85% |
| | 31–36 | 5,26% | 1,58% | 1,72% | 3,85% |
| | Older than 36 | 2,02% | 2,08% | 0,25% | 1,10% |
| University | Private | 85,81% | 0,89% | 1,56% | 0,55% |
| | Public | 14,19% | 99,11% | 98,44% | 99,45% |
| Field | Arts and Humanities | 7,95% | 2,48% | 0,82% | 7,14% |
| | Physical Sciences | 0,00% | 1,39% | 1,64% | 3,30% |
| | Social and Behavioral Sciences | 14,31% | 13,86% | 14,94% | 2,75% |
| | Education | 4,46% | 8,42% | 2,87% | 1,65% |
| | Engineering, Manufacturing and Construction | 15,72% | 16,93% | 20,61% | 69,23% |
| | Business and Administration | 54,07% | 46,63% | 59,11% | 12,64% |
| | Health and Well-being | 3,49% | 10,30% | 0,00% | 3,30% |

The courses reported by students were classified into different fields in accordance with UNESCO's [11] classification.

assessed on the basis of the CFI, TLI, RMSEA, and SRMR indices [34]. The analysis was performed using R [35], with the psych [36] and lavaan packages [37].

After evidencing the construct dimension of the perception of remote learning quality, their values were assessed by using statistical descriptions and Pearson correlations between dimensions. Countries were compared using one-way ANOVA with Tukey's post-hoc comparison test and Bonferroni correction [31] besides exploring the joint effect of the dimensions with the linear regression models. As an additional component, the potential effect of the interaction between the dimensions of the construct of quality perception was explored using the Hayes [38] extension for SPSS.

## Results, discussion, conclusions

### Qualitative analysis

The focus groups were guided by the following question: How does emergency remote education affect the quality of lessons? Undergraduate students of all socioeconomic strata taking courses based on six areas of knowledge in Colombian HEIs analyzed the impact of ERT on quality on the basis of the following aspects:

**Technology gap and availability of technological equipment.** The technology gap, just like the situation of social and economic inequality, is a determining factor in the perception of lesson quality. The students evidenced that in the case of poor and emerging countries, where inequality is a common feature, the major problem of ERT is related to accessibility [8, 10]. Not all of the academic community has access to ERT, given that some people may lack devices or connectivity, and professors are not always in a better condition than students are [18]. The quality of lessons must be based on improving student conditions. At home, a young person may get distracted in several ways. Internet quality may be poor, study space may be inadequate, equipment available may be obsolete, family relations may be complex, and there might even be family violence. Therefore, if the goal was to provide a solution to the continuity of presence-based education by means of ERT, the primary task was improving students' conditions and not just the cold delivery of contents.

With all these limitations, the students affirmed that learning is reduced in ERT because of the following reasons. i) The student receives classes in an inadequate learning environment. ii) Because of connectivity problems, some content is missing. iii) Professors and students do not properly use communication platforms. iv) Most professors and students are not adequately trained to bring about learning by means of these tools.

**Quality of lessons.** Undergraduate students consider that as part of the efforts to continue to provide the education service, universities have not considered the different variables adequately; therefore, the quality of most lessons significantly, and as a consequence, students are concerned regarding how the university will guarantee the quality processes in a confinement situation.

Furthermore, students claim that the quality of remote lessons presents issues related to the following aspects. i) Professors have limited classes to conduct PowerPoint presentations. ii) Professors have focused on the need to transmit and on the need to adapt classes to the challenges of ERT. However, the students recognize that this is because of the short transition period from in-person classes to ERT. iii) There is no planning, which leads to an overflow of assignments with no feedback. iv) The improvisation of ERT has made professors deal with contents under the belief that the student is a recipient of information, thereby minimizing the importance of internal development and class practicality and downplaying everything to the theoretical plane. v) Remote education affected academic quality by reducing diverse teaching

contents, study time, and the development of practical capacities, which cannot be achieved to the same extent in a virtual environment.

The most serious impact is the loss in dialectical discussion agility caused by non-face-to-face-classes. The ERT process has limitations that discourage the confrontation of ideas. Active participation is limited, especially in subjects requiring practice and/or lab exercises that are impossible to carry out remotely. Additionally, the empathic bond necessary for the transfer of learning has been decimated, given that regardless of the professors' and students' efforts, the emotional connection is not the same in remote and in-person education. Despite the foregoing, if there is passion for learning, obstacles can be overcome and may even become promoters of innovations and new pedagogical findings.

**Pedagogical tools.** The major problem of ERT is related to pedagogical and didactic processes. In circumstances of confinement, not being present in the same premises as (isolation/seclusion from) the students and their classroom environment limits the implementation of knowledge transmission to just one way. This way, students who do not understand or do not have sufficient technological skills fail to acquire all the knowledge transmitted. Conversely, professors' training in the use of pedagogical strategies reveals the effort required to create learning conditions such as the opening of spaces for counseling and delivery of classroom material.

As a whole, the conclusion of this qualitative analysis is that undergraduate students find the quality of the lessons to be affected by ERT. Less-than-adequate Internet access and lack of technological devices (PCs, smartphones, etc.) are some of the academic community limitations. Additionally, the scant preparation, planning, and decrease of content affect the quality of learning processes. Finally, students recognize the efforts that professors put into pedagogical strategies.

## Quantitative analysis

The quantitative analysis allowed for the identification of three dimensions in the construct of ERT's quality of lessons as per parallel analysis. Additionally, the factorial weights allowed the retention of 20 items with weights over 0.4. Table 3 shows the factorial weights that result from EFA in each country.

Once the items were grouped as per EFA, the research group discussed the construct behind each dimension. Factors were labeled 'Concerns about educational quality' (CEQ), 'Teaching strategies adopted by the professor' (TSAP), and 'Perception of access limitations' (PAL). The first includes items showing the participants' worries with regard to the quality of the learning process because of the change from the presence-based modality to the online methodology. In this dimension, high scores represent a greater concern about the effect of ERT on quality of lessons. Furthermore, the TSAP factor highlights the professor's active role in planning and implementing the training activities; a high score represents a positive perception of the teaching strategies adopted. Finally, PAL evaluates the perception of the problems experienced by students when they tried to access the online class.

The factorial structure identified was independently tested in each country by means of CFA. Table 4 summarizes the adjustment indices of every country. All values show a suitable adjustment of the three-factor structure to data, even in Costa Rica, where EFA was not previously conducted.

Finally, the constructed measures' reliability was studied. CEQ and TSAP showed appropriate values across all countries. However, the PAL scale exceeds the reference value of 0.7 [31] in Mexico and Peru, while being slightly behind this value in Colombia and Costa Rica.

**Table 3. Exploratory factor analysis.**

| Item | Colombia | | | México | | | Perú | | |
|---|---|---|---|---|---|---|---|---|---|
| | F1 | F2 | F3 | F1 | F2 | F3 | F1 | F2 | F3 |
| The absence of a direct relationship between professors and students weakens the construction of analytical and critical thinking. | **0,66** | -0,09 | 0,05 | **0,72** | 0,02 | -0,05 | **0,64** | 0 | 0,04 |
| Remote lessons changed evaluations, involving impact on quality. | **0,74** | 0,01 | -0,09 | **0,63** | 0,03 | -0,04 | **0,67** | 0,05 | -0,01 |
| The quality of remote lessons is lower than that of in-person classes. | **0,84** | -0,03 | -0,05 | **0,78** | -0,06 | -0,08 | **0,76** | -0,04 | -0,06 |
| Remote lessons affect my educational quality. | **0,89** | -0,01 | -0,02 | **0,89** | 0 | 0,01 | **0,8** | -0,08 | -0,06 |
| I am concerned about how the university will guarantee quality throughout my learning process. | **0,8** | -0,06 | -0,07 | **0,65** | -0,01 | 0 | **0,69** | 0,11 | -0,06 |
| Remote lessons reduced the learning contents. | **0,64** | -0,08 | 0,09 | **0,72** | -0,07 | 0,08 | **0,65** | -0,11 | 0,11 |
| I can't concentrate during online lessons; there are many distractions. | **0,63** | 0,06 | 0,25 | **0,5** | 0,01 | 0,3 | **0,5** | 0,08 | 0,2 |
| The development of aptitudes and competences is limited with online education. | **0,62** | 0,01 | 0,08 | **0,68** | -0,04 | 0,03 | **0,6** | 0,03 | 0,1 |
| I have doubts about the quality of online education. | **0,77** | -0,05 | 0 | **0,69** | -0,07 | 0 | **0,64** | -0,07 | 0,05 |
| Online education discourages human relationships, which are vital to consolidate learning. | **0,7** | 0,15 | 0,07 | **0,64** | 0,07 | 0,04 | **0,51** | 0,05 | 0,07 |
| Professors encourage interaction in their online lessons. | -0,04 | **0,73** | 0,02 | -0,07 | **0,72** | 0,04 | -0,06 | **0,7** | 0,05 |
| Professors provide widely available supporting material. | 0,11 | **0,79** | -0,11 | 0,07 | **0,69** | -0,18 | 0,09 | **0,72** | -0,16 |
| Professors have excellent platform usage skills. | -0,08 | **0,71** | 0,01 | 0,05 | **0,8** | -0,05 | 0,02 | **0,62** | 0,04 |
| The lesson development is not disrupted by the professor's environment. | 0,07 | **0,53** | -0,12 | -0,01 | **0,61** | -0,06 | 0,06 | **0,61** | -0,05 |
| Professors incorporate new strategies, facilitating learning during online classes. | -0,11 | **0,73** | 0,07 | -0,07 | **0,81** | 0,12 | -0,08 | **0,73** | 0,06 |
| Professors open consultation spaces to clarify unanswered doubts during online lessons. | -0,07 | **0,55** | 0,04 | -0,08 | **0,69** | 0 | -0,11 | **0,56** | 0,1 |
| My knowledge of virtual platforms is limited. | 0,22 | 0,09 | **0,42** | 0,19 | 0,18 | **0,47** | 0,29 | 0,06 | **0,42** |
| My environment is not suitable for online classes. | 0,32 | -0,02 | **0,55** | 0,27 | 0,01 | **0,61** | 0,37 | 0,04 | **0,51** |
| The audio and/or camera of the computer I use is not working properly. | -0,04 | -0,02 | **0,76** | -0,09 | -0,09 | **0,78** | 0,02 | -0,06 | **0,73** |
| I must share my computer with my nuclear family, so I cannot always attend remote lessons. | -0,04 | -0,06 | **0,72** | 0 | -0,02 | **0,73** | -0,04 | 0,01 | **0,85** |

Table 5 shows the statistical description of the values measured. Scores were converted into a standardized normal score to facilitate interpretation. All correlations were statistically significant, and similar results were obtained in all countries. CEQ had a negative relationship with TSAP but a positive relationship with PAL. Participants indicate that in situations wherein they could perceive better planning and implementation of training activities by lecturers, their concerns about the educational quality have diminished. Nonetheless, the opposite happened when greater access constraints were suffered by them.

On the other hand, TSAP had a negative relationship with PAL. The lecturer's active role in activity planning and implementation has helped mitigate the perceived challenges to access education.

Colombia and Mexico have been the countries with the greatest concerns with regard to educational quality, followed by Costa Rica and Peru. Significant differences have been observed among countries $F(3.4041) = 37.64$, $p < 0.001$. Peru differs significantly from the rest as Peruvian students reported to have fewer concerns about educational quality than students from the remaining countries under study did. There are no differences between Colombia, Mexico, and Costa Rica.

**Table 4. Confirmatory factor analysis.**

| | Chi | Df | CFI | TLI | RMSEA | SRMR |
|---|---|---|---|---|---|---|
| Colombia | 512,17 | 167 | 0,989 | 0,987 | 0,05 | 0,05 |
| México | 432,805 | 167 | 0,989 | 0,987 | 0,056 | 0,055 |
| Perú | 531,589 | 167 | 0,984 | 0,982 | 0,06 | 0,054 |
| Costa Rica | 259,169 | 167 | 0,987 | 0,985 | 0,055 | 0,082 |

**Table 5. Descriptive analysis.**

| Country | Variable | Mean | Deviation | Alpha | Correlations | |
|---|---|---|---|---|---|---|
| | | | | | CEQ | TSAP |
| Colombia | CEQ | 0,11 | 1,01 | 0,90 | | |
| | TSAP | 0,11 | 0,95 | 0,79 | -0.339*** | |
| | PAL | -0,26 | 0,94 | 0,68 | 0,381*** | -0,232*** |
| Costa Rica | CEQ | 0,09 | 1,03 | 0,89 | | |
| | TSAP | -0,05 | 1,02 | 0,82 | -0,469*** | |
| | PAL | -0,51 | 0,87 | 0,61 | 0,414*** | -0,423*** |
| México | CEQ | 0,11 | 0,99 | 0,88 | | |
| | TSAP | -0,29 | 1,10 | 0,84 | -0,396*** | |
| | PAL | 0,04 | 1,01 | 0,71 | 0,367*** | -0,209*** |
| Perú | CEQ | -0,25 | 0,94 | 0,86 | | |
| | TSAP | 0,09 | 0,92 | 0,78 | -0,242*** | |
| | PAL | 0,38 | 0,95 | 0,73 | 0,528*** | -0,134*** |

All values are standardized with zero average and standard deviation 1. CEQ = Concerns about the educational quality; TSAP = Teaching strategies adopted by the professor; PAL = Perception of access limitations.

*** $p < .001$

Differences were also shown in the teaching strategies adopted by the professor, $F(3.4041)$ = 38.19, $p < 0.001$ across countries. In this case, Mexican participants were those reporting a negative perception of the teaching strategies implemented in class. No differences were observed between the remaining countries. Finally, differences were found in the perception of limitations, $F(3.4041)$ = 111.47, $p < 0.001$. Out of all the countries assessed, Peruvian students reported having the greatest level of perception of limitations, followed by Mexico, Colombia, and Costa Rica. All comparisons between countries were statistically significant as per the post-hoc tests, proving the differential degree of limitations across the countries under study.

Correlation analyses showed how the dimensions TSAP and PAL were significantly associated with CEQ, so the potential relationship between them was explored through a stepwise regression analysis. This analysis technique provides a more complete view of the demographic variables of the participants and the dimensions measured. Tables 6 and 7 show the results of the regression models, taking the score in the CEQ dimension as a dependent variable. Gender, age, university type, and knowledge were used as control variables in Model 1, and PAL and TSAP were added in Model 2. Analysis was carried out in each country. Dummy variables were constructed to add the variable 'field of knowledge.' As a final exploration, Model 3 was incorporated to study the potential effect of the interaction between PAL and TSAP.

Among the control variables, age was found to have a negative relationship with concerns about educational quality in all countries studied; older students had a lower perception of quality issues. The knowledge area of Education in Colombia and Peru was associated with a smaller number of concerns about educational quality, while Business in Mexico and Costa Rica was related to a larger number of concerns about quality.

When the TSAP and PAL dimensions were added, the analysis showed a negative relationship with TSAP and a positive relation with PAL. This result provides higher reliability on the presence of the said relationship as it is maintained despite the control variables accounting for a portion of model variance. Moreover, the dimensions of the quality perception construct explain the higher variance in accordance with the change in $R^2$.

**Table 6. Regression models in Colombia and México.**

| | Colombia | | | | | | México | | | | | |
| | Model 1 | | | Model 2 | | | Model 1 | | | Model 2 | | |
| Variable | B | *B* | SE | B | *B* | SE | B | *B* | SE | B | *B* | SE |
|---|---|---|---|---|---|---|---|---|---|---|---|---|
| Constant | 0,62 | | 0,34 | 0,71* | | 0,3 | 0,86 | | 1,19 | 2,06* | | 1,04 |
| Women | -0,18 | -0,09 | 0,32 | -0,03 | -0,01 | 0,28 | -0,47 | -0,23 | 0,99 | -1,33 | -0,66 | 0,86 |
| Men | -0,06 | -0,03 | 0,32 | 0,14 | 0,07 | 0,28 | -0,48 | -0,24 | 0,99 | -1,34 | -0,66 | 0,86 |
| Age | -0,22*** | -0,16 | 0,03 | -0,23*** | -0,17 | 0,03 | -0,13** | -0,11 | 0,04 | -0,10** | -0,09 | 0,03 |
| University | 0,15 | 0,05 | 0,08 | 0,06 | 0,02 | 0,07 | 0,05 | 0 | 0,33 | -0,18 | -0,02 | 0,29 |
| Art | 0,01 | 0 | 0,11 | 0,08 | 0,02 | 0,1 | 0,06 | 0,01 | 0,21 | 0,06 | 0,01 | 0,19 |
| Social Sciences | 0,04 | 0,01 | 0,09 | 0,02 | 0,01 | 0,08 | -0,14 | -0,05 | 0,12 | -0,17 | -0,06 | 0,1 |
| Education | -0,44** | -0,09 | 0,14 | -0,38** | -0,08 | 0,12 | 0,28* | 0,08 | 0,13 | 0,19 | 0,05 | 0,12 |
| Health | 0,22 | 0,04 | 0,15 | 0,21 | 0,04 | 0,13 | 0,02 | 0,01 | 0,12 | 0,02 | 0,01 | 0,11 |
| Administration | -0,11 | -0,05 | 0,07 | -0,12 | -0,06 | 0,06 | -0,18* | -0,09 | 0,09 | -0,25** | -0,12 | 0,08 |
| Physical Sciences | | | | | | | 0,07 | 0,01 | 0,27 | 0 | 0 | 0,24 |
| TSAP | | | | -0,28*** | -0,26 | 0,02 | | | | -0,29*** | -0,32 | 0,03 |
| PAL | | | | 0,36*** | 0,33 | 0,02 | | | | 0,31*** | 0,32 | 0,03 |
| $R^2$ | 0,05 | | | 0,26 | | | 0,03 | | | 0,27 | | |
| $\Delta R^2$ | | | | 0,22*** | | | | | | 0,24*** | | |

TSAP = Teaching strategies adopted by the professor; PAL = Perception of access limitations.

*$p < .05$,

**$p < .01$,

***$p < .001$

Finally, the PAL showed a moderating effect in the relationship between TSAP and CEQ, $\beta = .088$, p < .0001. The impact of the pedagogical strategies on the concerns about quality is negative, so the strategies adopted by the professor allow for a reduction in the students' worries regarding quality. However, this effect gets weaker as access limitations increase. In cases of great limitations, $\beta = .-018$, p < .0001, the relationship is weaker than those facing average access restrictions, $\beta = -.26$, p < .0001. In turn, the situations of average access limitations were found to have a weaker impact than the situations of greater limitations did, $\beta = -.35$, p < .0001. Fig 1 represents the moderating effect of access limitations.

## Discussion

The decision taken by universities to continue giving lessons through the pandemic in order to preserve the relationship with its main stakeholder, their students, requires a further assessment of its consequences regarding the quality of lessons provided and the students' perception as clients.

The study has explored the perception of students regarding the quality of service throughout the ERT period [5, 12]. To this end, it delved into the perception of students from four Latin American countries that adopted similar measures to sustain education in the pandemic time.

Students are concerned about the decreasing of quality of lessons. Students considered that universities had not conducted an analysis to find out the best way to adapt to ERT. The teaching strategy was limited to the professors' impromptu transfer of knowledge in a scarcely planned manner [27]. ERT opted for an education model based on the delivery of knowledge, giving no time for understanding, lacking a space open to develop practical skills and apply the

**Table 7. Regression models in Peru and Costa Rica.**

| | Perú | | | | | | Costa Rica | | | | | |
|---|---|---|---|---|---|---|---|---|---|---|---|---|
| | Model 1 | | | Model 2 | | | Model 1 | | | Model 2 | | |
| Variable | B | Beta | SE | B | Beta | SE | B | Beta | SE | B | Beta | SE |
| Constant | -0,27 | | 0,45 | 0,52 | | 0,38 | 1,87 | | 2,21 | 2,06 | | 1,95 |
| Women | | | | | | | 0,54 | 0,26 | 0,92 | 0,28 | 0,14 | 0,81 |
| Men | 0,1 | 0,05 | 0,06 | 0,14** | 0,07 | 0,05 | 0,94 | 0,45 | 0,92 | 0,6 | 0,29 | 0,81 |
| Age | -0,30*** | -0,14 | 0,06 | -0,30*** | -0,14 | 0,05 | -0,45** | -0,26 | 0,13 | -0,36** | -0,21 | 0,11 |
| University | 0,27 | 0,04 | 0,21 | -0,2 | -0,03 | 0,18 | -0,77 | -0,06 | 1,02 | -0,71 | -0,05 | 0,9 |
| Art | 0,72* | 0,07 | 0,3 | 0,3 | 0,03 | 0,25 | 0,26 | 0,07 | 0,27 | -0,08 | -0,02 | 0,24 |
| Social Sciences | 0,21* | 0,08 | 0,09 | 0,07 | 0,03 | 0,08 | -0,12 | -0,02 | 0,42 | 0,06 | 0,01 | 0,37 |
| Education | 0,46** | 0,08 | 0,17 | 0,26 | 0,05 | 0,14 | -0,65 | -0,08 | 0,53 | -0,76 | -0,09 | 0,47 |
| Health | | | | | | | -0,63 | -0,11 | 0,43 | -0,39 | -0,07 | 0,38 |
| Administration | 0,04 | 0,02 | 0,07 | 0,02 | 0,01 | 0,06 | -0,69** | -0,22 | 0,22 | -0,61** | -0,2 | 0,2 |
| Physical Sciences | -0,08 | -0,01 | 0,22 | -0,42* | -0,06 | 0,18 | 0,5 | 0,09 | 0,38 | 0,21 | 0,04 | 0,34 |
| TSAP | | | | -0,18*** | -0,17 | 0,02 | | | | -0,24*** | -0,24 | 0,07 |
| PAL | | | | 0,50*** | 0,51 | 0,02 | | | | 0,34*** | 0,29 | 0,08 |
| $R^2$ | 0,04 | | | 0,34 | | | 0,25 | | | 0,39 | | |
| $\Delta R^2$ | | | | 0,30*** | | | | | | 0,18*** | | |

TSAP = Teaching strategies adopted by the professor; PAL = Perception of access limitations.

\*$p < .05$,

\*\*$p < .01$,

\*\*\*$p < .001$

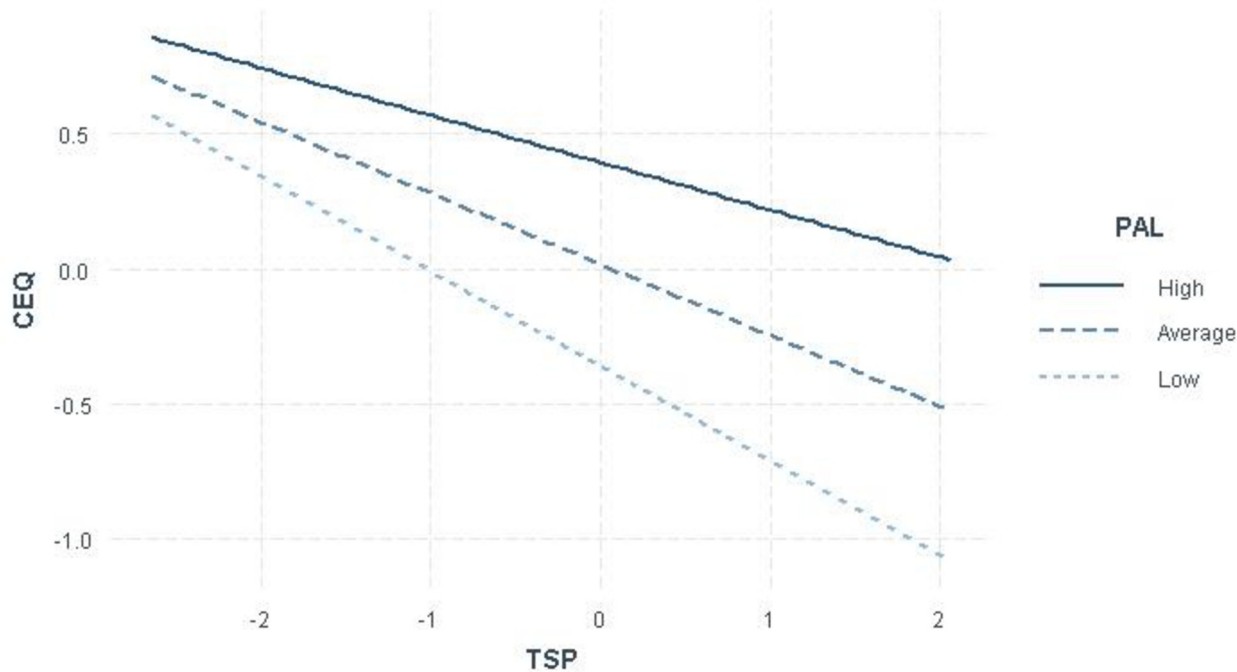

**Fig 1. The effect of the interaction of access limitations.** The figure represents the difference found in the relationship between teaching strategies and concerns about quality. TSAP = Teaching strategies adopted by the professor; PAL = Perception of access limitations.

concepts learned. ERT transformed the professor–student interaction, giving the latter a passive role as content recipient [28].

In the professor–student interaction, the former had a pivotal role in the learning process, requiring their additional effort to develop teaching strategies allowing for content understanding. Students acknowledge the effort that professors made although they consider that despite it, the structural limitations to access remote lessons hamper the teaching–learning process.

The social inequality conditions present in Latin America as described by Malleret & Schwab [4] were clearly reflected in the differential access to ERT. The technology gap described by Moreno [8] turned into a factor against the quality of lessons, which affected not only students but also professors, who had to deal with connectivity issues and the lack of resources to give classes. Likewise, low expertise in the use of technological tools led to an even greater impact on quality perception. In addition to the limited technological skills demonstrated by students, they considered that universities remained indifferent and did not provide training in technological tools management.

In short, students' perception of quality of lessons may be grouped into three dimensions as shown in the qualitative analyses, supported by the results of the quantitative analysis applied. Those three dimensions highlighted in this study involved concerns about educational quality, teaching strategies adopted by the professor, and perception of access limitations.

After comparing the countries under study, significant differences with regard to concerns about the quality of lessons were observed. Colombian students reported the highest level of concern in this dimension, but they acknowledged the greater effort made by professors in developing teaching strategies. Likewise, it is found to be one of the countries wherein students perceive a lower level of access limitations. For their part, just like Colombians, Costa Rican students expressed a similar level of concern about quality of lessons although they consider that the effort made to adopt teaching strategies was lower. In addition, Costa Rican students report the lowest perception of access limitations.

Mexican students stand out for their low perception of the didactic strategies implemented by professors. Mexico shows the worst values for this dimension. The level of concern about quality of lessons is similar to that of Colombian and Costa Rican students while access limitations had average values across the countries compared. Finally, Peruvian students have the lowest degree of concern about quality despite the highest level of perception of access limitations obtained. Just like Colombian students, they acknowledge the great effort made by professors to develop teaching strategies.

Quantitative analysis allowed for exploring the effect of students' demographic variables and the university's institutional variables with regard to concerns about quality of lessons. Older students reported fewer concerns about quality, which may be associated with the stage of the university course of studies they were at. More advanced students have developed knowledge and skills that help them better understand the content taught during ERT than students who were just beginning their professional course of studies at the beginning of the pandemic. Moreover, the differences found between the fields of knowledge were few. The Peruvian and Colombian students who were taking courses in the Education area reported to have a lower level of concern about quality, while the same effect was present across Business Administration students in Mexico and Costa Rica.

In view of these results, there is no solid differential effect between fields of knowledge although these may be distinguished by their degree of implementation. Furthermore, the institutional variable 'type of university' had no effect. Therefore, the quality perceived by students from public and private universities is similar.

The relationship between the constructed dimensions allowed the achievement of a noteworthy result. The didactic strategies helped reduce the students' concerns about quality of

lessons. During the student–professor interaction, described in the preceding paragraphs, the activity planning and implementation methods during lessons became tools to deal with the threats associated with quality lessons perception. As the qualitative analysis participants state, the professors' lack of training to handle remote teaching turned into the central component of the teaching–learning process, reversing the investment on education, which is student-centered. The said result is a point of agreement between the qualitative and quantitative analyses; hence, the role of the professor during ERT required a huge effort to develop didactic strategies in addition to the emotional supportive role described by Pérez et al. [7, 12].

In addition, this study could provide evidence on the moderating role of access limitations in the training process. The relationship identified between teaching strategies and concerns about the quality of lessons is modified by the degree of limitations perceived. As the moderation analysis shows, this relationship is stronger when there are fewer access limitations although it is weakened as they increase. Therefore, the teaching strategies by themselves were not capable of addressing the potential concerns about the quality of lessons. Therefore, the quality of lessons is limited by the access that students have to sustain the teaching–learning process.

This study opens some possibilities for future research. First, in this study, it has been confirmed that the construct of perception for students depends on the context where the measuring was carried out. Therefore, a comparison between countries with different characteristics, different educational systems, or different connectivity coverage would allow the expansion of knowledge on this construct. Second, the ability to measure student perception in times of a pandemic allows us to address other research questions related to the effectiveness of the quality of lessons and how HEIs adjust their quality of service before new crises. This last point in particular would imply considering other dimensions in the students' perception of perception during the pandemic as well as the strategic tactics related to client services.

An element that requires future research is the stability of the construct over time. Although in this study an assessment of student perception at the beginning of ERT has been carried out, it is necessary to replicate it after a year because the HEIs' responses may have been modified and thus generated changes in perception.

## Theoretical implications

This study makes several theoretical contributions. First, the dimension of the construct Quality of Service is shown during the ERT period. The lack of a construct's dimension hinders the development of future research because it is no longer possible to measure it or build knowledge based on it. In this article, it is shown that the construct quality of lessons in ERT is expressed based on three dimensions. This conceptual exercise provides the basis for deepening the functioning of the construct.

Once the construct is identified, it would be possible to study the relations among its dimensions, in addition to its relation with other constructs. Thus, a second theoretical implication consists of understanding the factors that led to a decrease in the quality of lessons perceived by the students in HEIs located in the countries under study. The task of the professor is a factor that stimulates quality perception and can be developed by the HEIs. Therefore, professor accompaniment and training represent a type of intervention that would finally affect students' perception of quality. However, issues related to access are structural elements related to the degree of development in each country, which is out of the control of HEIs. Although HEIs work on developing the skills of their professors, such an effect fails to change the structural deficiencies in the quality of lessons.

Finally, the construct's dimension and the way in which the relation of its dimensions is similar in developing countries are shown in this article. Although this should be carried out

in other countries, there is evidence to believe that the relation established between the three construct dimensions may be spread to countries having similar educational systems and connectivity.

## Practical implications

The decision of keeping the teaching process using ERT had consequences for the HEIs that have now been identified. As shown in this article, students' perception of quality of lessons decreased, but factors that account for it are also presented. As a first practical consequence, this article enables HEIs to understand what happened during ERT, which factors should be strengthened, and which ones need to be mitigated.

Second, the HEIs are presented with the impact that their decisions had on students. The ERT affected quality perception, and it was interpreted by students as a rushed and poorly reasoned decision by the HEIs. The teaching task was delivered to the professors, and the students felt indifference by the HEIs, which put the relationship with their main stakeholders at risk.

Finally, measuring the results of previously carried out analyses allows the building of a student profile as the client, in times of crisis, and this may be used to identify differentiating factors that contribute to HEIs' effective introduction into the market.

## Limitations

This study has several limitations. First, the qualitative analysis that helped develop the measurement instrument was only conducted in Colombia. Thus, focus groups from each of the countries under assessment could make additional comments to expand the contents of the constructed instrument. Conversely, the sampling performed was not randomized, which may restrict the generalizability of the results. In addition, as quantitative analyses are correlational, it is impossible to explain causality, which can only ever be offered regarding the relationship between results. Moreover, clarifications regarding the differences reported between countries cannot be provided, as this is an exploratory study.

## Acknowledgments

This research was possible by the support in the data collection process of Angel Antonio Fernández Montiel, Gina Paola Bravo, Alex Alberto Nicolls, Sandra Cristina Castro Becerra, Claudia Patricia Gómez Espinosa, Milkiades Guarín, William Farfán, Paula Andrea Valencia+, Paula Franco, Santiago Patarroyo, Manuel Darío Palacio, Javier Quiñones Quiroz, Alexander Restrepo, Ariel Camilo González, Gonzalo Alberto Sabogal, Karolina Urbano, and Gerardo Peña.

## Author Contributions

**Conceptualization:** Lida Esperanza Villa Castaño, William Fernando Durán, Paula Andrea Arohuanca Percca.

**Formal analysis:** Lida Esperanza Villa Castaño, William Fernando Durán, Paula Andrea Arohuanca Percca.

**Investigation:** Lida Esperanza Villa Castaño, William Fernando Durán, Paula Andrea Arohuanca Percca.

**Methodology:** Lida Esperanza Villa Castaño, William Fernando Durán.

**Project administration:** Lida Esperanza Villa Castaño, William Fernando Durán.

**Resources:** Lida Esperanza Villa Castaño.

**Validation:** William Fernando Durán.

**Writing – original draft:** Lida Esperanza Villa Castaño, William Fernando Durán, Paula Andrea Arohuanca Percca.

**Writing – review & editing:** Lida Esperanza Villa Castaño, William Fernando Durán.

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
