## [Decision Letter · Decision Letter 0]

2 Feb 2022

PONE-D-21-32218Remote teaching satisfaction within the context of COVID-19: Comparative study in Latin AmericaPLOS ONE

Dear Dr. Villa,

Thank you for submitting your manuscript to PLOS ONE. After careful consideration, we feel that it has merit but does not fully meet PLOS ONE’s publication criteria as it currently stands. Therefore, we invite you to submit a revised version of the manuscript that addresses the points raised during the review process.

Both reviewers are positive about the impact of the work, but have a number of specific concerns and suggestions that should be address point-by-point. 

We look forward to receiving your revised manuscript.

Kind regards,

Michael Klymkowsky, Ph.D.

Academic Editor

PLOS ONE

Journal Requirements:

3. During our internal checks, the in-house editorial staff noted that you conducted research or obtained samples in another country. Please check the relevant national regulations and laws applying to foreign researchers and state whether you obtained the required permits and approvals. Please address this in your ethics statement in both the manuscript and submission information. In addition, please ensure that you have suitably acknowledged the contributions of any local collaborators involved in this work in your authorship list and/or Acknowledgements. Authorship criteria is based on the International Committee of Medical Journal Editors (ICMJE) Uniform Requirements for Manuscripts Submitted to Biomedical Journals - for further information please see here: https://journals.plos.org/plosone/s/authorship.

Reviewers' comments:

Reviewer's Responses to Questions

**Comments to the Author**

1. Is the manuscript technically sound, and do the data support the conclusions?

Reviewer #1: Partly

Reviewer #2: Yes

2. Has the statistical analysis been performed appropriately and rigorously? 

Reviewer #1: Yes

Reviewer #2: Yes

3. Have the authors made all data underlying the findings in their manuscript fully available?

Reviewer #1: No

Reviewer #2: No

4. Is the manuscript presented in an intelligible fashion and written in standard English?

Reviewer #1: Yes

Reviewer #2: Yes

5. Review Comments to the Author

Reviewer #1: *3. Have the authors made all data underlying the findings in their manuscript fully available?

Ans.　I couldn't find the description. Added; I couldn't find any ethical state.

1)This is a valuable study that investigated students' perceptions of quality in the pandemic. There could be a pandemic going forward. Research results may be a measure of student status during a pandemic, rather than in peacetime. The clearer the extraction process for the question items, the more reliable it is.

Introduction, an overview of this study is included, but there is no presentation of the purpose of the research.

2) The analysis method is reasonable. However, the method of qualitative analysis, which is the basis of this study, is unclear.

It seems that there is a lack of analytical methods for parts qualitatively.

3) Why did only students from the area participate in Group Focus Interview (GFI)?　How did the GIF progress?

4) I couldn't find a description of how to recruit participants, GFI, and online survey.

5) Are there any exclusion criteria? Do you have ethics permission? about GFI, and online survey. I don't know the relationship between qualitative analysis and questions. The question is, where does it belong to GFI?

6)" Concerns about educational quality" is the equivalent of QS? The relationship between the title used "satisfaction", QS, and CEQ, should be explained for ease of understanding.

Why did you ask a question “how do you think ERT affects QS?”

Why didn't you extract an opinion about satisfaction from GFI?

Where did it replace.

7)Is "TSP" in the table same "TSAP" in the body?. The expression, if possible, is kind to match the body or table.

Reviewer #2: Dear Authors,

It is a well written article and a needed topic of the hour.

I have few suggestions to improve the manuscript. Methodology has to be improved by adding the details of quantitative part of the study.

The length of the article is can be reduced by removing the repeated sentences in many paragraphs.

Information about the questionnaires used in conducting the quantitative part of the study is missing.

Discussion can include similar studies from other countries as ERT was introduced all over the world because of COVID pandemic.

Thank you

6. PLOS authors have the option to publish the peer review history of their article (what does this mean?). If published, this will include your full peer review and any attached files.

Reviewer #1: No

Reviewer #2: **Yes: **Jansi Rani Natarajan

---

## [Author Response · Author response to Decision Letter 0]

20 Mar 2022

March 10, 2022

Michael Klymkowsky, Ph.D.

Academic Editor

PLOS ONE

Dr Dr Klymkowsky,

We would like to express our gratitude for reviewing the article titled “Remote teaching satisfaction within the context of COVID-19: Comparative study in Latin America”. We consider that the comments made have improved the quality of the article and we have integrated all adjustment recommendations. 

Following, we provide an answer to reviewer’s comments

1. Introduction

Evaluator: Introduction, an overview of this study is included, but there is no presentation of the purpose of the research.

Comment: We accept the comment. We have included the purpose of the research in the introduction. 

2. Method

Evaluator:

The analysis method is reasonable. However, the method of qualitative analysis, which is the basis of this study, is unclear.

It seems that there is a lack of analytical methods for parts qualitatively.

Why did only students from the area participate in Group Focus Interview (GFI)?　How did the GIF progress?

I couldn't find a description of how to recruit participants, GFI, and online survey. 

Are there any exclusion criteria? Do you have ethics permission? about GFI, and online survey. I don't know the relationship between qualitative analysis and questions. The question is, where does it belong to GFI?

Comment: We have described deeply the GFI, in the page 6. We post a massive invitation in the social media and participants respond to it message. All participants were voluntaries.

About the request of the area of participants, we think that the reviewer refer to the country where the GFI were applied. Effectively, GFI were performed in Colombia but, quantitative results support the similar expression of the measure in all the countries. As we show in the quantitative results, there are not differences in the construct among countries.

Finally, we clarify the relationship between qualitative analysis and quantitative questions. Qualitative findings enable to write the quantitative questions using the expressions of the participants and the dimensional structure drawn.

3. Discussion

Evaluator: 6)" Concerns about educational quality" is the equivalent of QS? The relationship between the title used "satisfaction", QS, and CEQ, should be explained for ease of understanding. 

Comment: We have adjusted the title of the article. Moreover, the comment made by the evaluator allowed us realize the inadequate use of the term “satisfaction” which we changed for “perception”. In addition, we have eliminated QS and replaced it with “quality of lessons” which adjusts better to the research.

Evaluator: Why did you ask a question “how do you think ERT affects QS?” Why didn't you extract an opinion about satisfaction from GFI? Where did it replace, " Concerns about educational quality"

Comment: In line with the comment before, we have changed the QS concept for quality of the lessons. With the new section in the methodology, we hope to give answer to your question.

Evaluator: 7) Is "TSP" in the table same "TSAP" in the body?　 The expression, if possible, is kind to match the body or table.

Comment: We had a typography error, which we have adjusted. 

Evaluator: Discussion can include similar studies from other countries as ERT was introduced all over the world because of COVID pandemic.

Comment: We included two additional studies concerning the region. 

Evaluator: The length of the article is can be reduced by removing the repeated sentences in many paragraphs.

Comment: We have made a thorough review of the article, and it has allowed us to submit a shorter version. 

We hope the revised version is now suitable for publication and look forward to hearing from you in due course.

Thank you for your consideration!

Sincerely,

Lida Esperanza Villa Castaño

Universidad Cooperativa de Colombia, Sede Bogotá

---

## [Decision Letter · Decision Letter 1]

12 May 2022

Remote teaching satisfaction within the context of COVID-19: Comparative study in Latin America

PONE-D-21-32218R1

Dear Dr. Villa,

We’re pleased to inform you that your manuscript has been judged scientifically suitable for publication and will be formally accepted for publication once it meets all outstanding technical requirements.

Kind regards,

Michael Klymkowsky, Ph.D.

Academic Editor

PLOS ONE

Additional Editor Comments (optional):

Reviewers' comments:

Reviewer's Responses to Questions

**Comments to the Author**

1. If the authors have adequately addressed your comments raised in a previous round of review and you feel that this manuscript is now acceptable for publication, you may indicate that here to bypass the “Comments to the Author” section, enter your conflict of interest statement in the “Confidential to Editor” section, and submit your "Accept" recommendation.

Reviewer #1: All comments have been addressed

Reviewer #2: All comments have been addressed

2. Is the manuscript technically sound, and do the data support the conclusions?

Reviewer #1: Yes

Reviewer #2: Yes

3. Has the statistical analysis been performed appropriately and rigorously? 

Reviewer #1: Yes

Reviewer #2: Yes

4. Have the authors made all data underlying the findings in their manuscript fully available?

Reviewer #1: Yes

Reviewer #2: Yes

5. Is the manuscript presented in an intelligible fashion and written in standard English?

Reviewer #1: Yes

Reviewer #2: Yes

6. Review Comments to the Author

Reviewer #1: (No Response)

Reviewer #2: Dear Authors,

Manuscript is improved to a better level.

I have one concern regarding the age group - Inclusion criteria says that age above 18 years, but table says age under 18 years as well. Please modify it.

English editing is still needed in some parts of the article.

7. PLOS authors have the option to publish the peer review history of their article (what does this mean?). If published, this will include your full peer review and any attached files.

Reviewer #1: No

Reviewer #2: **Yes: **Jansi Rani Natarajan

---

## [Editor Report · Acceptance letter]

19 May 2022

PONE-D-21-32218R1 

Perception of the Quality of Remote Lessons in the Time of COVID-19: A Comparative Study in Latin America Quality of remote lessons in the context of COVID-19 

Dear Dr. Villa:

I'm pleased to inform you that your manuscript has been deemed suitable for publication in PLOS ONE. Congratulations! Your manuscript is now with our production department. 

Kind regards, 

on behalf of

Dr. Michael Klymkowsky 

Academic Editor

PLOS ONE